# INTELLIGENT MATRIX EXPONENTIATION

## ABSTRACT

We present a novel machine learning architecture that uses a single high-dimensional nonlinearity consisting of the exponential of a single input-dependent matrix. The mathematical simplicity of this architecture allows a detailed analysis of its behaviour, providing robustness guarantees via Lipschitz bounds. Despite its simplicity, a single matrix exponential layer already provides universal approximation properties and can learn and extrapolate fundamental functions of the input, such as periodic structure or geometric invariants. This architecture outperforms other general-purpose architectures on benchmark problems, including CIFAR-10, using fewer parameters.

## 1 INTRODUCTION

Deep neural networks (DNNs) synthesize highly complex functions by composing a large number of neuronal units, each featuring a basic and usually 1-dimensional nonlinear activation function $f : \mathbb{R}^1 \rightarrow \mathbb{R}^1$. While highly successful in practice, this approach also has disadvantages. In a conventional DNN, any two activations only ever get combined through summation. This means that such a network requires an increasing number of parameters to approximate more complex functions even as simple as multiplication. Parameter-wise, this approach of composing simple functions does not scale efficiently.

An alternative to the composition of many 1-dimensional functions is using a simple higher-dimensional nonlinear function $f : \mathbb{R}^m \rightarrow \mathbb{R}^n$. A single multidimensional nonlinearity may be desirable because it could express more complex relationships between input features with potentially fewer parameters and fewer mathematical operations.

The matrix exponential stands out as a promising but overlooked candidate for a higher-dimensional nonlinearity for machine learning models. The matrix exponential is a smooth function that appears in the solution to one of the simplest differential equations that can yield desirable mathematical properties: $d/dt y(t) = My(t)$, with the solution $y(t) = \exp(Mt)y(0)$, where $M$ is a constant matrix. The matrix exponential plays a prominent role in the theory of Lie groups, an algebraic structure widely used throughout many branches of mathematics and science.

A unique advantage of the matrix exponential is its natural ability to represent oscillations and exponential decay, which becomes apparent if we decompose the matrix to be exponentiated into a symmetric and an antisymmetric component. The exponential of an antisymmetric matrix, whose nonzero eigenvalues are always imaginary, generates a superposition of periodic oscillations, whereas the exponential of a symmetric matrix, which has real eigenvalues, expresses an exponential growth or decay. This fact, especially the ability to represent periodic functions, gives the M-layer the possibility to extrapolate beyond its training domain. Many real-world phenomena contain some degree of periodicity and can therefore benefit from this feature. In contrast, the functions typically used in conventional DNNs approximate the target function locally and are therefore unable to scale well outside the boundaries of the training data for such problems.

Based on these insights, we propose a novel architecture for supervised learning whose core element is *a single layer* (henceforth referred to as "M-layer"), that computes *a single matrix exponential*, where the matrix to be exponentiated is an affine function of the input features. We show that the M-layer has universal approximator properties and allows closed-form per-example bounds for robustness. We demonstrate the ability of this architecture to learn multivariate polynomials, such as matrix determinants, and to generalize periodic functions beyond the domain of the input without any feature engineering. Furthermore, the M-layer achieves results comparable to recently-proposed

non-specialized architectures on image recognition datasets. We provide TensorFlow code that implements the M-layer in the Supplementary Material.

## 2 RELATED WORK

Neuronal units with more complex activation functions have been proposed. One such example are sigma-pi units (Rumelhart et al., 1986), whose activation function is the weighted sum of products of its inputs. More recently, neural arithmetic logic units have been introduced (Trask et al., 2018), which can combine inputs using multiple arithmetic operators and generalize outside the domain of the training data. In contrast with these architectures, the M-layer is not based on neuronal units with multiple inputs, but uses a single matrix exponential as its nonlinear mapping function. Through the matrix exponential, the M-layer can easily learn mathematical operations more complex than addition, but with simpler architecture. In fact, as shown in Section 3.3, the M-layer can be regarded as a generalized sigma-pi network with built-in architecture search, in the sense that it learns by itself which arithmetic graph should be used for the computation.

Architectures with higher-dimensional nonlinearities are also already used. The softmax function is an example of a widely-used such nonlinear activation function that, like the M-layer, has extra mathematical structure. For example, a permutation of the softmax inputs produces a corresponding permutation of the outputs. Maxout networks also act on multiple units and have been successful in combination with dropout (Goodfellow et al., 2013). In radial basis networks (Park & Sandberg, 1991), each hidden unit computes a nonlinear function of the distance between its own learned centroid and a single point represented by a vector of input coordinates. Capsule networks (Sabour et al., 2017) are another recent example of multidimensional nonlinearities. Similarly, the M-layer uses the matrix exponential as a single high-dimensional nonlinearity. This mapping satisfies nontrivial mathematical identities, for some of which we here elucidate their application potential in a machine learning context. Feature crosses of second and higher order have also been explored on top of convolutional layers ((Lin et al., 2015; Lin & Maji, 2017; Li et al., 2017; Engin et al., 2018; Koniusz et al., 2018). In this line of research, (Li et al., 2017) introduces the use of a matrix operation (either matrix logarithm or matrix square root) on top of a convolutional layer with higher order feature crosses. The M-Layer differs from this kind of approach in two ways: first, it uses matrix exponentiation to *produce* feature crosses, and not to improve trainability on top of other ones; secondly, it computes a matrix operation of an arbitrary matrix (not necessarily symmetric positive semidefinite), which allows more expressibility, as seen for example in Section 3.3.

The main differences between the M-layer and other architectures that can utilize feature-crosses are the M-layer's ability to also model non-polynomial dependencies (such as a cosine) and the built-in competition-for-total-complexity regularization, which we will explain in Section 3.3 through a "circuit breadboard" analogy.

Matrix exponentiation has a natural alternative interpretation in terms of an ordinary differential equation (ODE). As such, the M-layer can be compared to other novel ODE-related architectures, such as neural ordinary differential equations (NODE) (Chen et al., 2018) and their augmented extensions (ANODE) (Dupont et al., 2019). We discuss this in Section 3.6.

Existing approaches to certifying the robustness of neural networks can be split into two different categories. Some approaches (Peck et al., 2017) mathematically analyze a network layer by layer, providing bounds on the robustness of each layer, that then get multiplied together. This kind of approach tends to give fairly loose bounds, due to the inherent tightness loss from composing upper bounds. Other approaches (Singh et al., 2018; 2019) use abstract interpretation on the evaluation of the network to provide empirical robustness bounds. In contrast, using the fact that the M-layer architecture has a single layer, in Section 3.7 we obtain a direct bound on the robustness on the whole network by analyzing the explicit formulation of the computation.

## 3 ARCHITECTURE

We start this section by refreshing the definition of the matrix exponential. We then define the proposed M-layer architecture and explain its ability to learn particular functions such as polynomials and periodic functions. Finally, we provide closed-form per-example robustness guarantees.

### 3.1 MATRIX EXPONENTIATION

The exponential of a square matrix $M$ is defined as: $\exp(M) = \sum_{k=0}^{\infty} \frac{1}{k!} M^k$. The matrix power $M^k$ is defined inductively as $M^0 = I$, $M^{k+1} = M \cdot M^k$, using the associativity of the matrix product; it is not an element-wise matrix operation. Note that the expansion of $\exp(M)$ in Eq. (3.1) is finite for nilpotent matrices. A matrix $M$ is called *nilpotent* if there exists a positive integer $k$ such that $M^k = 0$. Strictly upper triangular matrices are a canonical example.

Multiple algorithms for computing the matrix exponential efficiently have been proposed (Moler & Van Loan, 2003). We provide an implementation of the M-layer that employs either the TensorFlow function `tf.linalg.expm`, which uses the scaling and squaring method combined with the Padé approximation (Higham, 2005), or a simpler and faster approximation of the matrix exponential that only squares a fixed number of times, using the formula $\exp(M) \approx (I + \frac{M}{2^k})^{2^k}$. This approximation follows from the equivalent definition of $\exp(M) = \lim_{n \to \infty} (I + \frac{M}{n})^n$, and has $O(kn^{1.5})$ time complexity, where $n$ is the total number of elements in $M$.

### 3.2 M-LAYER DEFINITION

At the core of the proposed architecture (Figure 1) is an M-layer that computes a *single* matrix exponential, where the matrix to be exponentiated is an affine function of all of the input features. In other words, an M-layer replaces *an entire stack of hidden layers* in a DNN.

We exemplify the architecture as applied to a standard image recognition dataset, but this formulation is applicable to any other type of problem by adapting the relevant input indices. In the following equations, generalized Einstein summation is performed over all right-hand side indices not seen on the left-hand side.

Consider an example input image, encoded as a 3-index array $X_{yxc}$, where $y$, $x$ and $c$ are the row index, column index and color channel index, respectively. The matrix $M$ to be exponentiated is obtained as follows, using the trainable parameters $\tilde{T}_{ajk}$, $\tilde{U}_{axyc}$ and $\tilde{B}_{jk}$:

Figure 1: M-layer architecture diagram.

$$M_{jk} = \tilde{B}_{jk} + \tilde{T}_{ajk} \tilde{U}_{ayxc} X_{yxc} \tag{1}$$

First, $X$ is projected linearly to a $d$-dimensional latent feature embedding space by $\tilde{U}_{ayxc}$. Then, the 3-index tensor $\tilde{T}_{ajk}$ maps each such latent feature to an $n \times n$ matrix. Finally, a bias matrix $\tilde{B}_{jk}$ is added to the feature-weighted sum of matrices. The result is a matrix indexed by row and column indices $j$ and $k$. In order to allow for exponentiation, $M$ must be a square matrix. It is possible to contract the tensors $\tilde{T}$ and $\tilde{U}$ in order to simplify the architecture formula, but partial tensor factorization provides regularization by reducing the parameter count. An output $p_m$ is obtained as follows, using the trainable parameters $\tilde{S}_{mjk}$ and $\tilde{V}_m$:

$$p_m = \tilde{V}_m + \tilde{S}_{mjk} \exp(M)_{jk} \tag{2}$$

The matrix $\exp(M)$, indexed by row and column indices $j$ and $k$ in the same way as $M$, is projected linearly by the 3-index tensor $\tilde{S}_{mjk}$, to obtain a $h$-dimensional output vector. The bias-vector $\tilde{V}_m$ turns this linear mapping into an affine mapping. The resulting vector may be interpreted as accumulated per-class evidence and may then be mapped to a vector of probabilities via softmax.

Training can thus be done conventionally, by minimizing a loss function such as the $L_2$ norm or the cross-entropy with softmax, using backpropagation through matrix exponentiation. In general, there is no simple analytical expression for the derivative of $\exp(M)$, but one can instead compute derivatives for each step of the algorithm chosen to compute the exponential. For example, `tf.linalg.expm` uses the scaling-and-squaring algorithm with a Padé approximation, whose steps are all differentiable. We discuss the topic in more detail in Appendix A.2.

The nonlinearity of the M-layer architecture is provided by the $\mathbb{R}^d \to \mathbb{R}^h$ mapping $v \mapsto \tilde{V}_m + \tilde{S}_{mjk} \exp(M)_{jk}$. The count of trainable parameters of this component is $dn^2 + n^2 + n^2h + h$. This

count comes from summing the dimensions of $\tilde{T}_{ajk}$, $\tilde{B}_{jk}$, $\tilde{S}_{mjk}$, and $\tilde{V}_m$, respectively. We note that this architecture has some redundancy in its parameters, as one can freely multiply the $T$ and $U$ tensors by a $d \times d$ real matrix and, respectively, its inverse, while preserving the computed function. Similarly, it is possible to multiply each of the $n \times n$ parts of the tensors $\tilde{T}$ and $\tilde{S}$, as well as $B$, by both an $n \times n$ matrix and its inverse. In other words, any pair of real invertible matrices of sizes $d \times d$ and $n \times n$ can be used to produce a new parametrization that still computes the same function.

### 3.3 FEATURE CROSSES AND UNIVERSAL APPROXIMATION

A key property of the M-layer is its ability to generate arbitrary exponential-polynomial combinations of the input features. For classification problems, M-layer architectures are a superset of multivariate polynomial classifiers, where the matrix size constrains the complexity of the polynomial while at the same time not uniformly constraining its degree. In other words, simple multivariate polynomials of high degree compete against complex multivariate polynomials of low degree.

We provide a universal approximator proof for the M-layer in Appendix A.1, which relies on its ability to express any multivariate polynomial in the input features if a sufficiently large matrix size is used. We provide here an example that illustrates how feature crosses can be generated through the matrix exponential. Consider a dataset with the feature vector $(\phi_0, \phi_1, \phi_2)$ given by the $\tilde{U} \cdot x$ tensor contraction, where the relevant quantities for the final classification of an example are assumed to be $\phi_0$, $\phi_1$, $\phi_2$, $\phi_0\phi_1$, and $\phi_1\phi_2^2$. To learn this dataset, we look for an exponentiated matrix that makes precisely these quantities available to be weighted by the trainable tensor $\tilde{S}$. To do this, we define three $7 \times 7$ matrices $T_{0jk}$, $T_{1jk}$, and $T_{2jk}$ as $T_{001} = T_{102} = T_{203} = 1$, $T_{024} = T_{225} = 2$, $T_{256} = 3$, and 0 otherwise. We then define the matrix $M$ as:

$$M = \phi_0 T_0 + \phi_1 T_1 + \phi_2 T_2 = \begin{pmatrix} 0 & \phi_0 & \phi_1 & \phi_2 & 0 & 0 & 0 \\ 0 & 0 & 0 & 0 & 0 & 0 & 0 \\ 0 & 0 & 0 & 0 & 2\phi_0 & 2\phi_2 & 0 \\ 0 & 0 & 0 & 0 & 0 & 0 & 0 \\ 0 & 0 & 0 & 0 & 0 & 0 & 0 \\ 0 & 0 & 0 & 0 & 0 & 0 & 3\phi_2 \\ 0 & 0 & 0 & 0 & 0 & 0 & 0 \end{pmatrix}$$

Note that $M$ is nilpotent, as $M^4 = 0$. Therefore, we obtain the following matrix exponential, which contains the desired quantities in its leading row:

$$\exp(M) = I + M + \frac{1}{2}M^2 + \frac{1}{6}M^3 = \begin{pmatrix} 1 & \phi_0 & \phi_1 & \phi_2 & \phi_0\phi_1 & \phi_1\phi_2 & \phi_1\phi_2^2 \\ 0 & 1 & 0 & 0 & 0 & 0 & 0 \\ 0 & 0 & 1 & 0 & 2\phi_0 & 2\phi_2 & 3\phi_2^2 \\ 0 & 0 & 0 & 1 & 0 & 0 & 0 \\ 0 & 0 & 0 & 0 & 1 & 0 & 0 \\ 0 & 0 & 0 & 0 & 0 & 1 & 3\phi_2 \\ 0 & 0 & 0 & 0 & 0 & 0 & 1 \end{pmatrix}$$

The same technique can be employed to encode any polynomial in the input features using a $n \times n$ matrix, where $n$ is one unit larger than the total number of features plus the intermediate and final products that need to be computed. The matrix size can be seen as regulating the total capacity of the model for computing different feature crosses. With this intuition, one can read the matrix as a "circuit breadboard" for wiring up arbitrary polynomials.

### 3.4 FEATURE PERIODICITY

While the M-layer can express a wide range of functions using the exponential of nilpotent matrices, non-nilpotent matrices also have important applications, such as learning the periodicity of input features. This is a problem where conventional DNNs struggle, as they cannot naturally generalize beyond the distribution of the training data. To illustrate how matrix exponentials can naturally fit periodic dependency on input features, without requiring an explicit specification of the periodic nature of the data, consider the matrix $M_r = \begin{pmatrix} 0 & -\omega \\ \omega & 0 \end{pmatrix}$. We have $\exp(tM_r) = \begin{pmatrix} \cos \omega t & -\sin \omega t \\ \sin \omega t & \cos \omega t \end{pmatrix}$, which is a 2d rotation by an angle of $\omega t$ and thus periodic in $t$ with period $2\pi/\omega$. This setup can fit functions that have an arbitrary period. Moreover, this representation of periodicity naturally extrapolates well when going beyond the range of the initial numerical data.

### 3.5 CONNECTION TO LIE GROUPS

The M-layer has a natural connection to Lie groups. Lie groups can be thought of as a model of continuous symmetries of a system such as rotations. There is a large body of mathematical theory

and tools available to study the structure and properties of Lie groups (Gilmore, 2008; 2012), which may ultimately also help for model interpretability.

Every Lie group has associated a Lie algebra, which can be understood as the space of the small perturbations with which it is possible to generate the elements of the Lie group. As an example, the set of rotations of 3-dimensional space forms a Lie group; the corresponding algebra can be understood as the set of rotation axes in 3 dimensions. Lie groups and algebras can be represented using matrices, and by computing a matrix exponential one can map elements of the algebra to elements of the group.

In the M-layer architecture, the role of the 3-index tensor $\tilde{T}$ is to form a matrix whose entries are affine functions of the input features. The matrices that compose $\tilde{T}$ can be thought of as generators of a Lie algebra. Building $M$ corresponds to selecting a Lie algebra element. Matrix exponentiation then computes the corresponding Lie group element. As rotations are periodic and one of the simplest forms of continuous symmetries, this perspective is useful for understanding the ability of the M-layer to learn periodicity in input features.

The connection to Lie groups is important because it can provide techniques for the regularization of the M-layer. It is, for example, possible to impose a constraint on the maximal depth of feature crosses by taking a generator-matrix tensor $\tilde{T}$ that itself is a tensor product with a nilpotent matrix algebra. Other Lie-group-inspired regularization techniques will be discussed in future work.

### 3.6 DYNAMICAL SYSTEMS INTERPRETATION

Recent work has proposed a dynamical systems interpretation of some DNN architectures. The NODE architecture (Chen et al., 2018) uses a nonlinear and not time-invariant ODE that is provided by trainable neural units, and computes the time evolution of a vector that is constructed from the input features. This section discusses a similar interpretation of the M-layer.

Consider an M-layer with $\tilde{T}$ defined as $\tilde{T}_{012} = \tilde{T}_{120} = \tilde{T}_{201} = +1$, $\tilde{T}_{210} = \tilde{T}_{102} = \tilde{T}_{021} = -1$, and 0 otherwise, with $\tilde{U}$ as the $3 \times 3$ identity matrix, and with $\tilde{B} = 0$. Given an input vector $a$, the corresponding matrix $M$ is then $\begin{pmatrix} 0 & a_2 & -a_1 \\ -a_2 & 0 & a_0 \\ a_1 & -a_0 & 0 \end{pmatrix}$. Plugging $M$ into the linear and time invariant (LTI) ODE $d/dt\, Y(t) = MY(t)$, we can observe that the ODE describes a rotation around the axis defined by $a$. Moreover, a solution to this ODE is given by $Y(t) = \exp(tM)Y(0)$. Thus, by choosing $S_{mjk} = Y(0)_k$ if $m = j$ and 0 otherwise, the above M-layer can be understood as applying a rotation with input dependent angular velocity to some basis vector over a unit time interval.

More generally, we can consider the input features to provide affine parameters that define a time-invariant linear ODE, and the output of the M-layer to be an affine function of a vector that has evolved under the ODE over a unit time interval. In contrast, the NODE architecture uses a non-linear ODE that is not input dependent, which gets applied to an input-dependent feature vector.

### 3.7 CERTIFIED ROBUSTNESS

We show that the M-layer allows a novel proof technique to produce closed-form expressions for guaranteed robustness bounds. For any matrix norm $\|\cdot\|$, we have (Roger & Charles, 1994):

$$\|\exp(X + Y) - \exp(X)\| \leq \|Y\| \exp(\|Y\|) \exp(\|X\|)$$

We also make use of the fact that $\|M\|_F \leq \sqrt{n}\|M\|_2$ for any $n \times n$ matrix, where $\|\cdot\|_F$ is the Frobenius norm and $\|\cdot\|_2$ is the 2-norm of a matrix. We recall that the Frobenius norm of a matrix is equivalent to the 2-norm of the vector formed from the matrix entries.

Let $M$ be the matrix to be exponentiated corresponding to a given input example $x$, and let $M'$ be the deviation to this matrix that corresponds to an input deviation of $\tilde{x}$, i.e. $M + M'$ is the matrix corresponding to input example $x + \tilde{x}$. Given that the mapping between $x$ and $M$ is linear, there is a per-model constant $\delta_{in}$ such that $\|M'\|_2 \leq \delta_{in}\|\tilde{x}\|_\infty$. The 2-norm of the difference between the outputs can be bound as follows:

$$\|\Delta_o\|_2 \leq \|S\|_2 \|\exp(M + M') - \exp(M)\|_F \leq \sqrt{n}\|S\|_2 \|\exp(M + M') - \exp(M)\|_2 \leq$$
$$\leq \sqrt{n}\|S\|_2\|M'\|_2 \exp(\|M'\|_2)\exp(\|M\|_2) \leq \sqrt{n}\|S\|_2\delta_{in}\|\tilde{x}\|_\infty \exp(\delta_{in}\|\tilde{x}\|_\infty)\exp(\|M\|_2)$$

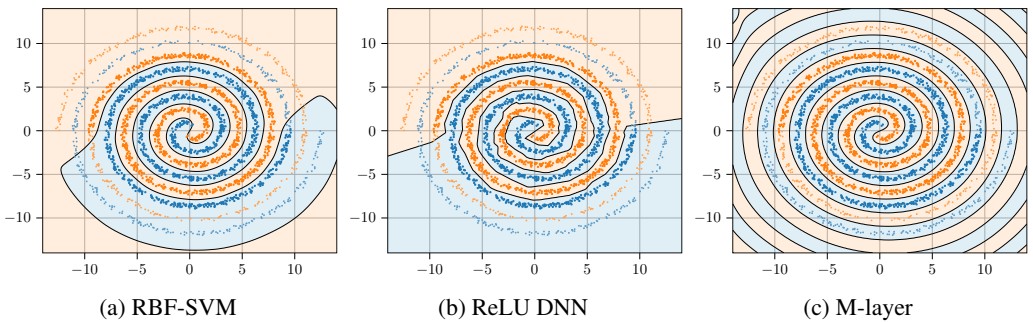

|               |                |               |
| :-----------: | :------------: | :-----------: |
|   (a) RBF-SVM |   (b) ReLU DNN |   (c) M-layer |

Figure 2: Classification boundaries, training and test sets on a "Swiss roll" classification task.

where $\|S\|_2$ is computed by considering $S$ a $h \times n \cdot n$ rectangular matrix, and the first inequality follows from the fact that the tensor multiplication by $S$ can be considered a matrix-vector multiplication between $S$ and the result of matrix exponential seen as a $n \cdot n$ vector.

This inequality allows to compute the minimal $L_\infty$ change required in the input given the difference between the amount of accumulated evidence between the most likely class and other classes. Moreover, considering that $\|x\|_\infty$ is bounded from above, for example by $1$ in the case of CIFAR-10, we can obtain a Lipschitz bound by replacing the $\exp(\delta_{in}\|\tilde{x}\|_\infty)$ term with a $\exp(\delta_{in})$ term.

An alternative way of studying robustness for a specific example involves the diagonalization of the matrix that corresponds to that example. In particular, we can use the Baker-Campbell-Hausdorff (BCH) formula to understand the neighborhood of the example. For example, if we regularize the tensor $T$ to punish non-commutativity of generator matrices, we expect the terms in the BCH formula to decrease in magnitude according to the number of commutators, so we can obtain an understanding of the non-linear effects in the neighborhood of each specific example. To our knowledge, no equivalent type of analysis exists for ReLU networks.

## 4 RESULTS

In this section, we show the performance of the M-layer on multiple benchmarks, in comparison with more traditional architectures. For each experiment, the hyperparameters are chosen based on performance. The DNNs use uniform Glorot initialization (Glorot & Bengio, 2010). The M-layers are initialized using normal distributions ($\mu = 0$) for the spiral, periodicity and determinants experiments ($\sigma = \{1, 0.01, 1\}$, respectively), and uniformly in $[-0.05, 0.05]$ for the image experiments. A regularization term is used on the M-layer for the spiral ($L_2 \cdot 10^{-3}$), determinants ($L_2 \cdot 10^{-4}$) and image ($L_1 \cdot 10^{-3}$) experiments. The selected number of epochs, batch size and learning rate are: $1000, 16, 10^{-2}$ (spiral); $1000, 128/16$ (M-layer/ReLU), $10^{-5/-4}$ (periodicity); $256, 32, 10^{-3}$ (determinants); $150, 32, 10^{-3}$ (images). The learning rate is reduced on plateaus and early stopping is used in all experiments except periodicity. RMSprop optimizers are used for all experiments except for images, where SGD with momentum $0.9$ is used. The loss function is the cross-entropy with softmax (spiral, images) or the mean squared error (MSE) (periodicity, determinants). We also used support-vector machines (SVM) with Gaussian radial basis function (RBF) as implemented by Pedregosa et al. (2011) as a baseline. RBF-SVM can be regarded as a neural network with one hidden layer. Unless otherwise stated, we report the best performance across SVM parameters $C = \{1, 10, 100\}$ and $\gamma = \{1/(\text{n\_features} \cdot \text{input\_variance}), 1/\text{n\_features}\}$.

### 4.1 LEARNING DOUBLE SPIRALS

To study the extrapolation capability of the classification boundaries generated by the M-layer, we compare it to a ReLU-DNN and a RBF-SVM on a double spiral ("Swiss roll") classification task. The data consist of 2000 randomly generated points along two spirals consisting of 3 rotations with coordinates in the $[-10, 10]$ range. Uniform random noise in the $[-0.5, 0.5]$ range is added to each input coordinate. As we are interested in extrapolation beyond the training domain, we generate a test set containing of the continuation of those spirals, wrapping around the origin one more times.

The M-layer has a representation size $d = 10$ and a matrix size $n = 10$. The DNN has two hidden layers of size 20. We report the best result out of 5 runs. On the test set, the DNN has an accuracy of 54%, while the M-Layer achieves an accuracy of 100%. The best RBF-SVM has an accuracy of 52%; similar SVM experiments can be found in (Suykens, 2001). Figure 2 illustrates the distinctive ability of the M-layer to extrapolate beyond the training domain.

## 4.2 LEARNING PERIODIC DATA

We compare an M-layer, DNNs and RBF-SVM on a dataset containing the daily minimum temperatures in Melbourne recorded by the Australian Bureau of Meteorology over 10 years (Datamarket.com, as cited by Kaggle.com). The data is de-meaned and smoothed using 7-day windows. The training set consists of the first 9 years, while the final year is used for testing. We report the best result out of 5 runs.

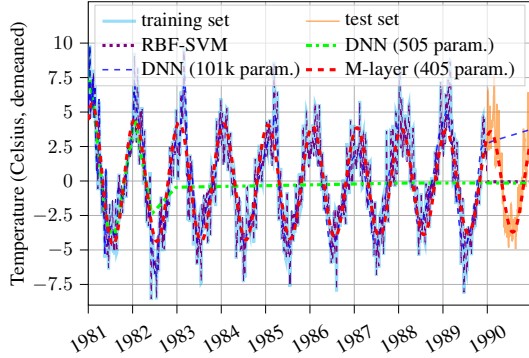

Figure 3: Predictions on a periodic dataset.

The M-layer has $d = 2$ and $n = 10$, resulting in a trainable parameter count of 405. During the initialization of the M-layer, a value of 10 is subtracted from the matrix diagonal in order to make it more likely for the initial matrix to be exponentiated to have negative eigenvalues and therefore keep outputs small. The first ReLU DNN has two hidden layers of size 20 each, resulting in a trainable parameter count of 505. The second ReLU DNN has a linear layer followed by 10 layers of size 100, resulting in a trainable parameter count of 101301, and is trained for 5000 epochs (a minimual configuration that was able to fit the training data). As the aim of this experiment is to show the ability to learn the periodicity of the input without additional engineering, we do not consider DNNs with special activation functions such as $\sin(x)$.

Figure 3 shows the predictions made by the trained M-layer, DNNs and RBF-SVM. The M-layer is able to pick up the periodic signal, whereas the other models attempt to fit the small-scale structure of the data, but are unable to extrapolate. Moreover, the DNN with comparable number of parameters to the M-layer does not have the capacity to learn the whole training set. The RBF-SVM is able to fit the training data in high detail, but, as expected, is nonetheless unable to extrapolate beyond the training domain. During training, we observe that the M-layer first searches for an approximate solution in a high error range, then rapidly decreases the loss once a suitable periodic approximation is found. We also observe that the results are sensitive to parameter changes. One viable solution to this, which we have explored but defer to future work, is to add an extra parameter controlling the ratio of symmetric and antisymmetric components of the matrix to train. The eigenvalues of an antisymmetric matrix are always imaginary or zero, meaning that the matrix exponential will always expresses oscillations, whereas the symmetric component will express exponential growth or decay (see also Section 3.4).

## 4.3 LEARNING DETERMINANTS

To demonstrate the ability of the M-layer to learn polynomials, we train M-layers, DNNs and RBF-SVMs to predict the determinant of $3 \times 3$ and $5 \times 5$ matrices. Learning the determinant of a matrix with a small network is challenging due to the size of the search space: a $5 \times 5$ determinant is a polynomial with 120 monomials of degree 5 in 25 variables; the generic inhomogeneous polynomial of this degree has $\binom{25+5}{5} = 142506$ monomials. We do not explicitly encode any special property of the determinant. From Section 3.3, we know that it is possible to express this multivariate polynomial with a single strictly upper-triangular and therefore nilpotent matrix; here we demonstrate that an unconstrained M-layer is equally capable of learning polynomials.

The data consist of $n \times n$ matrices with entries sampled uniformly between $-1$ and 1. With this sampling, the expected value of the square of the determinant is $\frac{n!}{3^n}$. So, we expect the square of the determinant to be $\frac{2}{9}$ for a $3 \times 3$ matrix, and $\frac{40}{81}$ for a $5 \times 5$ matrix. This means that an estimator

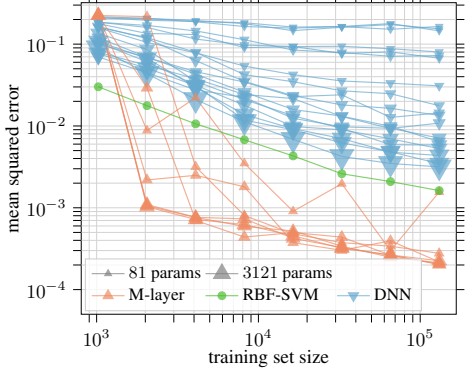

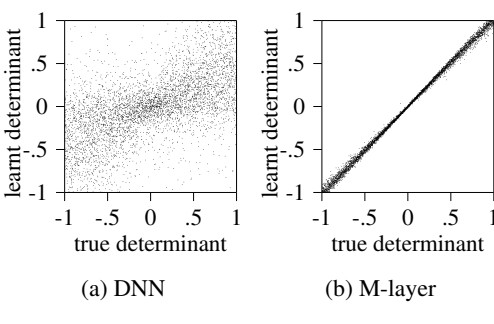

(a) DNN        (b) M-layer

Figure 4: Learning the determinant of a $3 \times 3$ matrix with various hyperparameters.

Figure 5: Learning the $5 \times 5$ determinant. Each scatter plot shows 5000 points.

constantly guessing 0 would have a MSE of $\approx 0.2222$ and $\approx 0.4938$ for the two matrix sizes, respectively, which provides an error baseline for the results. The size of the training set is between $2^{10}$ and $2^{17}$ examples for the $3 \times 3$ matrices, and $2^{20}$ for the $5 \times 5$ matrices. The validation set is $25\%$ of the training set size, in addition to it. Test sets consist of $10^6$ matrices.

The M-layer has $d = 9$ and $n$ between 6 and 12 for $3 \times 3$ determinants, and $n = 24$ for $5 \times 5$ determinants. The DNNs has 2 to 4 equally-sized hidden layers, each consisting of $5, 10, 15, 20, 25$ or 30 neurons, for the $3 \times 3$ matrices, and 5 hidden layers of size 100 for the $5 \times 5$ matrices.

Figure 4 shows the results of learning the determinant of $3 \times 3$ matrices. The M-layer architecture is able to learn from fewer examples compared to the DNN. The best M-layer model learning on $2^{17}$ examples achieves a MSE of $\approx 2 \cdot 10^{-4}$ with 811 parameters, while the best DNN has a MSE of $\approx 0.003$ with 3121 parameters. Figure 5 shows the results of learning the determinant of $5 \times 5$ matrices. An M-layer with 14977 parameters outperformed a DNN with 43101 parameters, achieving a MSE of 0.279 compared to 0.0012. We observe equivalent results for the matrix permanent.

Table 1: Comparison of classification performance on image recognition datasets. We compare the M-layer to fully-connected ReLU nets, RBF-SVM, NODE (Chen et al., 2018), and ANODE (Dupont et al., 2019). The ReLU net has 2 hidden layers of size (100, 50) for MNIST and (50, 50) for CIFAR-10 and SVHN. For NODE and ANODE, we report numbers from Dupont et al. (2019).

| ARCHITECTURE | CONVOLUTIONAL? | PROBLEM | ACCURACY % (MEAN ± S.D.) | PARAMETERS |
|---|---|---|---|---|
| M-LAYER | NO | MNIST | 97.99 ± 0.12 | 68 885 |
| RELU (F.C.) | NO | MNIST | 97.58 ± 0.10 | 84 060 |
| RBF-SVM | NO | MNIST | 98.37 | |
| NODE | YES | MNIST | 96.40 ± 0.50 | 84 000 |
| ANODE | YES | MNIST | 98.20 ± 0.10 | 84 000 |
| M-LAYER | NO | CIFAR-10 | 54.17 ± 0.36 | 148 965 |
| RELU (F.C.) | NO | CIFAR-10 | 51.07 ± 0.62 | 156 710 |
| RBF-SVM | NO | CIFAR-10 | 56.86 | |
| NODE | YES | CIFAR-10 | 53.70 ± 0.20 | 172 000 |
| ANODE | YES | CIFAR-10 | 60.60 ± 0.40 | 172 000 |
| M-LAYER | NO | SVHN | 81.19 ± 0.23 | 148 965 |
| RELU (F.C.) | NO | SVHN | 77.83 ± 1.23 | 156 710 |
| RBF-SVM | NO | SVHN | 74.45 | |
| NODE | YES | SVHN | 81.00 ± 0.60 | 172 000 |
| ANODE | YES | SVHN | 83.50 ± 0.50 | 172 000 |

### 4.4 LEARNING IMAGE DATASETS

We train M-layers on three image classification tasks: MNIST (Lecun et al., 1998), CIFAR10 (Krizhevsky, 2009), and SVHN (Netzer et al., 2011). The training set is shuffled and 10% of it is used for validation. Here, for RBF-SVM only, due to the long training times, we run the classification on CIFAR10 and SVHN only with the parameters that performed best on MNIST. The M-layer dimensions are always $d = 35$ and $n = 30$. The M-layer can be combined with convolutional layers for better performance, but we limit the scope of this work to the basics of the M-layer architecture. All accuracy values are averaged over at least 30 runs.

We compare the performance of the M-layer with three general-purpose architectures. As the M-layer is a novel architecture and no additional engineering is performed to obtain the results in addition to the regularization process described above, we only compare it to other generic architectures. As shown in Table 1, the M-layer outperforms the fully-connected ReLU network and the convolutional NODE architecture while using fewer parameters. Only ANODE, an improved version of NODE, outperforms the M-layer for all datasets. RBF-SVM performs better than all other models on MNIST and worse than all other models on SVHN.

To assess the performance of computing a matrix exponential, we recorded training times on the CIFAR-10 task on an NVIDIA V100 GPU. The training time per epoch for the M-layer is $5.28s \pm 0.34$ using the `tf.linalg.expm` implementation and $2.8s \pm 0.18$ using the fast approximation described in Section 3.1 with $k = 3$, compared to $2.41s \pm 0.05$ for the ReLU network, where all models are trained with batch size 32. The approximation using $k \geq 3$ does not affect the accuracy.

We also compute the robustness bounds of the M-layer trained on CIFAR-10, as described in Section 3.7. We train $n = 20$ models with $\delta_{in} \approx 200$, $\|S\|_2 \approx 3$, and $\|M\|_2$ typically a value between 3 and 4. The maximum $L_2$ variation of the vector of accumulated evidences is $\approx 1$. This results in a typical $L_\infty$ bound for robustness of $\approx 10^{-5}$ on the whole set of correctly classified CIFAR-10 test samples. In comparison, an analytical approach to robustness similar to ours (Peck et al., 2017), which uses a layer-by-layer analysis of a traditional DNN, achieves $L_2$ bounds of $\approx 10^{-9}$. Figure 6 shows the distribution of $L_\infty$ bounds obtained for the M-layer.

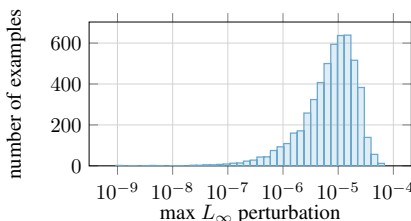

Figure 6: Maximum $L_\infty$ perturbation on the correctly classified CIFAR-10 test samples that is guaranteed not to produce a misclassification.

## 5 CONCLUSION

This paper introduces a novel model for supervised machine learning based on a single matrix exponential, where the matrix to be exponentiated depends linearly on the input. The M-layer is a powerful yet mathematically elegant architecture that has universal approximator properties and that can be used to learn and extrapolate several problems that traditional DNNs have difficulty with.

The M-layer architecture has a natural ability to learn input feature crosses, multivariate polynomials and periodic functions. This allows it to extrapolate learning to domains outside the training data with no additional engineering. Its performance is similar to that of non-specialized architectures, with competitive training times. Thanks to its powerful but relatively simple mathematical structure, the M-layer also allows closed-form robustness bounds.

We provide TensorFlow source code for the M-layer in the from of a Keras layer.

### 5.1 FUTURE WORK

In future work, we will focus on larger experiments involving advanced regularization techniques inspired by the connection between the M-layer and Lie groups. We will also show how the M-layer can be combined with convolutional layers into hybrid architectures for image recognition and regression.

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

# A APPENDIX

## A.1 UNIVERSAL APPROXIMATION THEOREM

We show that a single M-layer model that uses sufficiently large matrix size is able to express any polynomial in the input features. This is true even when we restrict the matrix to be exponentiated to be nilpotent or, more specifically, strictly upper triangular. So, for classification problems, M-layer architectures are a superset of multivariate polynomial classifiers, where matrix size constrains the complexity of the polynomial.

**Theorem 1** (Expressibility of polynomials). *Given a polynomial $p(x_1, \ldots, x_n)$ in $n$ variables, we can choose weight tensors for the M-layer such that it computes $p$ exactly.*

*Proof.* The tensor contraction applied to the result of matrix exponentiation can form arbitrary linear combinations, and is therefore able to compute any polynomial given a matrix that contains the constituent monomials up to constant factors. Thus, it suffices to prove that we can produce arbitrary monomials in the exponentiated matrix.

Given a monomial $m$ of degree $d-1$, we consider the $d \times d$ matrix $U$, that has the $d-1$ (possibly repeated) factors of the monomial on the first upper diagonal and zeros elsewhere. Let us consider powers of $U$. It can be shown that all elements of $U^i$ are equal to 0, except for the $i$-th upper diagonal, and that the value in the $(d-1)$-th upper diagonal of $U^{d-1}$, which contains only one element, is the product of the entries of the first upper diagonal of $U$. This is precisely the monomial $m$ we started with. By the definition of the exponential of the matrix, $\exp(U)$ then contains $\frac{m}{(d-1)!}$, which is the monomial up a constant factor.

Given a polynomial $p$ consiting of $t$ monomials, for each $1 \leq i \leq t$, we form matrices $U_i$ for the corresponding monomial $m_i$ of $p$, as described above. Then we build the diagonal block matrix $U = \mathrm{diag}(U_1, \ldots, U_t)$. It is clear that $\exp(U) = \mathrm{diag}(\exp(U_1), \ldots, \exp(U_k))$, so we can find all monomials of $p$ in $\exp(U)$. $\qquad \square$

To illustrate the proof, we look at a monomial $m = abcd$.

$$
U \quad = \quad \begin{pmatrix} 0 & a & 0 & 0 & 0 \\ 0 & 0 & b & 0 & 0 \\ 0 & 0 & 0 & c & 0 \\ 0 & 0 & 0 & 0 & d \\ 0 & 0 & 0 & 0 & 0 \end{pmatrix}
$$

$$
\exp(U) \quad = \quad \begin{pmatrix} 1 & a & \frac{1}{2}ab & \frac{1}{6}abc & \frac{1}{24}abcd \\ 0 & 1 & b & \frac{1}{2}bc & \frac{1}{6}bcd \\ 0 & 0 & 1 & c & \frac{1}{2}cd \\ 0 & 0 & 0 & 1 & d \\ 0 & 0 & 0 & 0 & 1 \end{pmatrix}
$$

The M-layers constructed here only make use of nilpotent matrices. When using this property as a constraint, the size of the M-layer can be effectively halved in the implementation.

The construction from Theorem 1 can be adapted to express not only a multivariate polynomial, i.e. a function to $\mathbb{R}^1$, but also functions to $\mathbb{R}^k$, which restrict to a polynomial in each coordinate. This, together with the Stone-Weierstrass theorem (Stone, 1948), implies the following:

**Corollary 2.** *For any continuous function $f \colon [a, b]^n \to \mathbb{R}^m$ and any $\epsilon > 0$, there exists an M-layer model that computes a function $g$ such that $|f(x_0, \ldots, x_{n-1})_j - g(x_0, \ldots, x_{n-1})_j| < \epsilon$ for all $0 \leq j \leq m$.*

### A.1.1 OPTIMALITY OF CONSTRUCTION

While our proof is constructive, we make no claim that the size of the matrix used in the proof is optimal and cannot be decreased. Given a multivariate polynomial of degree $d$ with $t$ monomials, the size of the matrix we construct would be $t(d+1)^2$. In fact, by slightly adapting the construction, we can obtain a size of matrix that is $td^2 + 1$. Given that the total number of monomials in polynomials of $n$ variables up to degree $d$ is $\binom{n+d}{d}$, it seems likely possible to construct much smaller M-layers for many polynomials. Thus, one wonders what is, for a given polynomial, the minimum matrix size to represent it with an M-layer.

As an example, we look at the determinant of a $3 \times 3$-Matrix. If the matrix is

$$
\begin{pmatrix} a & b & c \\ d & e & f \\ g & h & i \end{pmatrix},
$$

then the determinant is the polynomial $aei - afh - bdi + bfg + cdh - ceg$. From Theorem 1, we know that it is possible to express this polynomial perfectly with a single M-layer. However, already

an M-layer of size $8$ is sufficient to represent the determinant of a $3 \times 3$ matrix: If

$$
M = \begin{pmatrix}
0 & 0 & i & f & 0 & 0 & 0 & 0 \\
0 & 0 & h & e & 0 & 0 & 0 & 0 \\
0 & 0 & 0 & 0 & 2d & -2f & 0 & 0 \\
0 & 0 & 0 & 0 & -2g & 2i & 0 & 0 \\
0 & 0 & 0 & 0 & 0 & 0 & 3c & -3b \\
0 & 0 & 0 & 0 & 0 & 0 & 3a & 0 \\
0 & 0 & 0 & 0 & 0 & 0 & 0 & 0 \\
0 & 0 & 0 & 0 & 0 & 0 & 0 & 0
\end{pmatrix}
\tag{3}
$$

then $\exp(M)$ is

$$
\begin{pmatrix}
1 & 0 & i & f & -fg+di & 0 & -cfg+cdi & bfg-bdi \\
0 & 1 & h & e & -eg+dh & -fh+ei & -ceg+aei+cdh-afh & beg-bdh \\
0 & 0 & 1 & 0 & 2d & -2f & 3cd-3af & -3bd \\
0 & 0 & 0 & 1 & -2g & 2i & -3cg+3ai & 3bg \\
0 & 0 & 0 & 0 & 1 & 0 & 3c & -3b \\
0 & 0 & 0 & 0 & 0 & 1 & 3a & 0 \\
0 & 0 & 0 & 0 & 0 & 0 & 1 & 0 \\
0 & 0 & 0 & 0 & 0 & 0 & 0 & 1
\end{pmatrix},
$$

the sum of $\exp(M)_{0,7}$ and $\exp(M)_{1,6}$ is exactly this determinant. The permanent of a $3 \times 3$ matrix can be computed with an almost identical matrix, by removing all minus signs.

### A.2 GRADIENT BACKPROPAGATION THROUGH A MATRIX EXPONENTIAL

As shown by Speelpenning (1980) in his PhD thesis, which first formalized reverse mode automatic differentiation as an algorithm, we can backpropagate through any computational algorithm that satisfies some basic constraints. The computation needs to be executed in such a way that all intermediate quantities are retained alongside zero-initialized accumulators. These accumulators will, on the backward pass, collect contributions of the sensitivity of the end result with respect to the associated intermediate quantity. The total effort needed for the computation of the full gradient then is a bounded small integer multiple (typically around 5, and never above 8) of the effort needed to compute the function.

A basic efficient numerical algorithm that computes accurate matrix exponentials and is backpropagation-friendly in this sense uses these steps:

1. Compute some norm, such as the 2-norm, of the elements of the input matrix $M_0$ (on finite dimensional vector spaces, all norms are topologically equivalent).

2. Find an integer $k$ such that $(1/2^k) \cdot M_0$ has norm smaller than some threshold $N$ such that matrices with norm not exceeding this threshold can be computed with reasonable accuracy by evaluating some matrix rational function.

3. Compute $Q_0 := \mathrm{expm}(M_0/2^k)$ by using this matrix rational function. A simple choice here would be the Taylor polynomial, $\mathrm{expm}(M) \approx I + M + (1/2!)MM + (1/3!)MMM + \dots$, truncated at some moderate power. (Many numerical implementations of matrix exponentiation instead use a Padé Approximation here.)

4. Compute the sequence '$Q_0, Q_1 := Q_0 Q_0, Q_2 := Q_1 Q_1, \dots, Q_k = Q_{k-1} Q_{k-1}$ (i.e. square $k$ times, retaining intermediate results). Then, $\mathrm{expm} M_0 \approx Q_k$, due to $Q_k = \mathrm{expm}(M_0/2^k)^{2 \cdot 2 \cdots 2}$.

As all intermediate quantities are retained, backpropagation is straightforward. If uses Taylor approximation for small-norm matrices, the only steps involved are re-scalings and matrix addition and multiplications. With the Padé approximation, which uses a matrix rational function, there also is an effective matrix inversion step, i.e. the need to backpropagate through a linear solver.

Notably, the number of matrix multiplications executed by this algorithm is input data dependent, and in the context of this work, this enables the M-layer architecture to fit oscillatory data even for a large number of oscillations. If one instead were to e.g. approximate matrix exponentiation with a Taylor series alone, this would easily become infeasible. The reason for this can be most

easily seen when trying to compute $Y := \sin(310 \approx 2\pi \cdot 50)$ by Taylor-expanding the matrix exponential $\mathrm{expm}(xG)$ for

$$G := \begin{pmatrix} 0 & -1 \\ +1 & 0 \end{pmatrix}.$$

For $x = 310$, the $x^n/n!$ terms in the (alternating) Taylor series become $< 0.01$ only as late as $n \geq 843$, whereas ten halving-steps and Taylor expansion to only $n = 5$ gives us $\mathrm{expm}(xG)_{1,0} \approx 0.8509$ (whereas $\sin 310 \approx 0.8519$).

This data-adjustable-depth property is what allows the M-Layer to successfully fit and extrapolate oscillatory signals over many oscillations.

