# OpenReview forum: "Intelligent Matrix Exponentiation"
_ICLR.cc/2021/Conference — Reject_

### Official Review · AnonReviewer1 · 2020-10-29
**Lacks rigour!**

**Rating:** 4
**Confidence:** 4

**Review:**

This paper explores matrix exponentiation as an alternative non-linearity in a neural network. The key idea is to compute an affine transform of the inputs, there by generating an nxn feature map, followed by applying a matrix exponential to this feature map, that is subsequently used for classification. The paper provides several scenarios where matrix exponential could be an interesting non-linearity to use. Experiments are provided on synthetic examples, and standard image recognition benchmarks in a limited setting and show some promise.

Pros:
1. The idea of using matrix exponential as a non-linearity is an interesting idea that is perhaps not explored in the context of neural networks so far.
2. The paper provides several contexts in which the matrix exponential can be a good non-linearity to adopt.
3. Experiments on out-of-sample extrapolation shows promising results.

Cons:
1. I think the key factor missing in the paper is perhaps a lack of a solid motivation for using the matrix exponential. What does such an operator capture intuitively in a neural network that is not possible with prior activations, for example, tanh, elu, etc.?

2. Note that the matrix exponentiation operator has a complexity of O(n^1.5) for a feature matrix of nxn, and demands significant computational expense. For large n, which is what is typically used in deep networks, this operation could be impossible. It is unclear if this operation can be done on a GPU and if so to what approximation accuracy?

3. There are other matrix operations that have been attempted before; for example, matrix log-maps, which can also offer non-linearities, and have been used in bilinear CNN models for fine-grained recognition tasks. The paper may include these topics in the survey and contrast the technical advantages of using the exponential instead.

4. The experiments are very limited, and networks that use convolutional layers are shown to significantly outperform the proposed non-linearity.

5. The paper also lacks details in analyzing the feature maps produced by the non-linearity, or provide any gradient derivations, but relies on standard autograd packages for such. It is also not clear why the paper decided to use a full image as input, and why not use convolutional layers within the pipeline. The paper title is also a bit ambiguous as it is unclear what is "intelligent" regarding matrix exponentiation?

Overall, this paper explores an interesting non-linearity, however lacks theoretical or empirical rigour in showing that the approach is practically useful.

---------------------------------
Post-Rebuttal: Thanks to the authors for the detailed response. I think the idea has potential, however (resonating with the comments of other reviewers) I still think the paper should be significantly improved for better motivation, stronger theoretical results, and more elaborate and diverse experiments that can demonstrate the effectiveness of the method.

---

> ### Author Response · Authors · 2020-11-13
> **We firmly challenge the idea that this paper lacks rigour.**
>
> We thank the reviewer for the valuable comments. However, we firmly challenge the idea that our paper lacks rigour. On the contrary, we have explained in detail the M-layer architecture, its mathematical properties and why it is capable of better extrapolation for periodic and geometric functions compared to conventional DNNs, we provided universal approximation and certified robustness proofs, and we demonstrated its performance on both generic ML datasets (image classification) and on datasets that highlight its advantages over conventional DNNs.
>
> It is important to convey that the aim of this paper is to provide a thorough mathematical introduction to a new model, as opposed to delving into large and harder-to-interpret experiments without a solid theoretical foundation. We chose to use a significant part of the paper to explain the properties of the M-layer, and we performed experiments that were relatively simple but clear, showcasing the abilities of the model without leaving the possibility of confounding factors, as might have been the case with larger networks or datasets. This is why we chose not to present results combining the M-layer and convolutional layers for this paper, although we do aim to do this in future research. To clarify this, we have added a section on “Future Work”. Overall, especially considering that low interpretability is currently a problem in ML, we believe that this is the proper and mathematically rigorous way of introducing a novel architecture.
>
> A point-by-point response to all the questions raised by the reviewer will follow within the next few days.

---

> > ### Author Response · Authors · 2020-11-19
> > **We respond to each point raised by the reviewer.**
> >
> > 1. There are multiple advantages of the matrix exponential operator, which we have underlined throughout the paper. First, this operator can capture functions and combinations of the input features in a more effective way than conventional DNNs, resulting in fewer required parameters for similar performance. Secondly, the matrix exponential allows extrapolation beyond the training domain because it can naturally express oscillations and exponential decay functions, whereas activation functions like ReLU, tanh, ELU are unable to do so. Finally, this paper originates from a series of unrelated research that extensively used backpropagation through matrix exponentials (to which we cannot refer without revealing our identity), where we found that this is numerically more efficient than anticipated, and so we decided to show that matrix exponentiation is a useful tool in ML as well.
> >
> > 2. In our experience, the ability of the M-layer to fit complex data rapidly grows with increasing matrix size. An M-layer used as part of a larger (possibly DNN) architecture would not automatically require large matrix sizes. In a different line of research, we have used automatic differentiation on exponentiation of matrices of size up to 1000 x 1000, with acceptable performance. Finally, as reported in Section 4.4, training can be indeed done on the GPU, with training times comparable to those of a ReLU DNN.
> >
> > 3. Thank you for pointing out this research. We now discuss it in Section 2. Specifically, we have now emphasized that "the M-Layer differs from this approach in two ways: first, it uses matrix exponentiation to *produce* feature crosses, and not to improve trainability on top of existing ones; secondly, it computes a matrix operation of an arbitrary matrix (not necessarily symmetric positive semidefinite), which allows more expressibility, as explained in Section 3.3".
> >
> > 4. We strongly challenge the idea that the experiments are "very limited", in particular in the light of the page count constraint. We have performed four different types of experiments on a total of seven datasets. First, we did an experiment on "Swiss roll" synthetic data, showcasing the unique capability of the M-layer to extrapolate beyond the training domain. Secondly, we demonstrated the ability of the M-layer to extrapolate periodicity on a real-world dataset, which a ReLU DNN cannot do without additional feature engineering; moreover, a DNN with a similar number of parameters cannot even learn the training set. Then, we have shown the ability of the M-layer to outperform, with fewer parameters, ReLU DNNs on the problem of determinants and permanents, which can be useful for larger problems where higher-level features such as the volume are important (but where a more complex experiment would not clearly show what causes the performance improvement). Finally, we have compared the M-layer on generic benchmarks, where we do not expect it to shine, but rather to demonstrate its validity as a generic ML architecture comparable to other *non-specialised* architectures. Note that the architecture that outperforms the M-layer is already specialized through convolution.
> >
> > As a general remark, the scientific method tries to explain relevant differences in the simplest possible setting where they arise, where all distracting elements have been eliminated. As a comparison from physics, the dynamics of the Lorenz attractor was found by trimming down a 12-parameter atmospheric physics ordinary differential equation to the bare minimum that shows the observed puzzling behavior; the resulting 3-parameter model admits an understandable interpretation of the phenomenon that would be hard to discuss in the “more realistic” model. We are doing the same for our architecture here - presenting the differences in the simplest possible setting where they arise.
> >
> > 5. We have now addressed the question about gradient derivations in Section 2.2: "In general, there is no simple analytical expression for the derivative of $exp(M)$, but one can instead compute derivatives for each step of the algorithm chosen to compute the exponential. For example, tf.linalg.expm uses the scaling-and-squaring algorithm with a Padè approximation, whose steps are all differentiable". We have also added an appendix with further details. Regarding feature maps, these are usually used to understand deep CNNs, and were not of particular interest in this work, but can be explored in the future. We have explained above our reasoning about not including experiments with convolutions; we aimed to showcase the ability of the M-layer to extrapolate periodic and geometric functions with a small number of parameters, as opposed to presenting it as an architecture specialized on image classification. Finally, regarding the title, we clarify that we chose it as a concise way to refer to an architecture that *learns* (a common meaning of “intelligence” in ML and AI) using matrix exponentiation.

---

### Official Review · AnonReviewer3 · 2020-10-30
**Interesting approach, but some doubts regarding comparisons**

**Rating:** 5
**Confidence:** 3

**Review:**

### Summary

In this work, the authors propose the M-Layer, a neural network "layer" that consists in affinely transforming the input to a matrix, applying its matrix exponential, and applying another affine transformation. The parameters describing the affine transformations are the learnable parameters.
The authors provide a "universal approximation result" that shows that when using a sufficiently large latent space, the matrix exponential is able to replicate arbitrary polynomial functions of the input data, as well as periodic functions.
In numerical experiments on synthetic data (determinant of a matrix, swiss roll), time series data, as well as image data (MNIST, CIFAR10, SVHN), a single M-Layer compares favorably to neural ODEs as well as a very simplistic RELU network, while being outperformed by augmented neural ODE's.
The authors furthermore provide robustness certificates for the M-layer, and point out its connections to Lie-theory and dynamical systems interpretations of deep learning.

### Decision

On the positive I think that the idea of going beyond component-wise nonlinearities is interesting and the matrix exponential, with its rich mathematical structure, makes for an intriguing candidate. The authors analyse this proposal from many different view including robustness, approximation capability, computational implementation, and empirical results.

On the negative side, I find the comparison to conventional neural networks not entirely fair. In particular, the claims of improved representational power compared to traditional neural network layers seem exagerated and vague.
The connections drawn to Lie theory and dynamical systems seem superficial in that they mostly restate properties of the matrix exponential instead of providing interpretations of the M layer.
Finally, the numerical experiments are comparing a single M-layer to a very shallow dense RELU network. While I appreciate the intention of the authors to keep the experimental setup simple, comparing the performance of individual layers is a very artificial situation that does not seem to give any insight if M-layers are useful as layers for deep learning.
If they are not, however, this casts doubt on the usefulness and relevance of M-layers in general.

While I do think that the idea of an M-layer is interesting and in particular the relative stability bounds are promising, I feel that the message of the paper is diminished by the unclear comparison to existing deep learning methods.
This makes it difficult to assess the relevance of the work which is why, for now, I tend to recommend rejection.


### Detailed comments

#### Conceptual comparison to RELU:

"While highly successful in practice, this approach also has disadvantages. In a conventional DNN, any two activations only ever get combined through summation. This means that such a network requires an increasing number of parameters to approximate more complex functions even as simple as multiplication. This approach of composing simple functions does not generalize well outside the boundaries of the training data."

I would argue that this is misleading, since the results in later layers of a deep neural network depend on the activations in a highly nonlinear way. Furthermore, I believe there exist universal approximation results even for shallow neural networks with arbitrary width, just like the universal approximation results provided for M-layers are true only in the limit of infinitely large matrix exponentials.  They also do not provide any evidence for their claim that the lack of generalization is linked to the composing of simple functions. The authors do provide some experiments to this end on swiss roll and meteorological datasets. However these examples seem somewhat taylor-made for the exponential map approximation. Furthermore, the RELU network used is not even able to fit the training data well, which suggests this issue might be more due to a lack of model capacity.

#### Lie Algebras and dynamical systems

My understanding is that these sections mostly recapitulate how the matrix exponential appears in lie theory and the solution of linear ODE. However, I don't think they provide a compelling interpretation of the M-layer in terms of either of those points of view. In neural ODE for instance the input data to the layer can be thought of as a state of a dynamical system that is determined by the model weights.
In the M-layer, instead, the input becomes part of the system matrix, then the propagator of this system matrix (the exponential map) is computed, and finally this propagator is applied to initial conditions given by another set of weights by the M-layer.
I cannot think of a meaningful interpretation and I don't think the authors have provided one, either.

#### Numerical experiments

Reiterating what I wrote earlier, I really think that the performance of deep neural networks constructed with M-Layers needs to be compared to that of conventional deep neural networks to establish the relevance of the M-layer as a neural network layer.
In a nutshell: if the differences between M-layers and RELUs were to disappear when working with deeper networks, then why should one still use the M-layer.

=====================================================================================

After reading the rebuttal and the other reviews I still lean towards rejection, the main reason being that I am not convinced at this point that exponential nonlinearities are a direction worth pursuing.
I furthermore find the passages on dynamical systems and lie groups to be still lacking, for the same reasons as detailed above.

---

> ### Author Response · Authors · 2020-11-13
> **We clarify the comparison between the M-layer and ReLU networks.**
>
> [Part 1 of 2]
>
> We thank the reviewer for the valuable comments on the comparison between the M-layer and ReLU networks.
>
> We understand that this comparison came across as misleading, possibly because the space constraints limited some of the explanations. To clarify this comparison, we have now improved the Introduction with a point about parameter efficiency and added a paragraph on the advantage of the matrix exponential over widely-used activation functions like ReLU that approximate the target function locally. We do agree that the results in later layers of a deep neural network depend on the activations in a highly nonlinear way, and that there exist universal approximation results for shallow neural networks. Indeed, we know from the Kolmogorov-Arnold theorem that combining different features via adding non-linear activations is sufficient to represent any function to any given accuracy. However, this is not parameter-efficient: in a setting where the features only get combined via addition, something as simple as multiplying two features requires learning an approximate logarithm twice, and an approximate exponential, per every instance of where multiplication is useful. Indeed, throughout the paper, we provide evidence that the M-layer is more parameter-efficient than conventional ReLU networks.
>
> We have also addressed the reviewer’s concern regarding the lack of capacity of the ReLU network in the second experiment by adding to Figure 3 the results for a larger ReLU network, which has appropriate capacity to learn the training set. As shown, this larger network is still not able to extrapolate periodicity. This is another important aspect we would like to emphasize: unlike the M-layer, a ReLU network with no additional feature engineering will be unable to extrapolate oscillatory behavior, or even simple exponential decay, beyond that it has seen in its training examples. The ReLU network can only learn oscillatory behaviour by locally approximating small pieces of the function, which it will not learn anything about outside the training set. In contrast, we show (see section 3.4) that the M-layer can naturally represent oscillations and exponential decay through the matrix exponential operator. In addition to mathematical applications, many natural events are periodic and can therefore benefit from the extrapolation properties of the M-layer. We have added a new paragraph to the Introduction further elaborating on this.
>
> We would also like to add a clarification regarding the problems we employed for showcasing the extrapolation abilities of the M-layer. While these may be perceived as tailored to the M-layer, they can also be regarded as benchmarks for a model’s capacity of extrapolation on periodic and geometric datasets. We argue that the community might not be accustomed to this type of problems precisely because conventional DNNs do not perform well on them. With our experiments showing that an alternative architecture can have natural extrapolation abilities with relatively fewer parameters and comparable performance, we hope to persuade the community that such problems are also important as benchmarks.

---

> > ### Author Response · Authors · 2020-11-13
> > **We clarify the connection to Lie groups and the choice of experiments.**
> >
> > [Part 2 of 2]
> >
> > We thank the reviewer for the valuable comments on the connection to Lie groups and the numerical experiments.
> >
> > The second concern of the reviewer relates to the connection to Lie algebras, which is not considered strong enough. In fact, already thinking about the M-layer as a "circuit breadboard for feature crosses with built-in regularization as different circuits compete for space" is rooted in our research experience with nilpotent Lie algebras and provided the original intuition why such an architecture would be interesting to characterize. For the special case of the generator matrices forming a nilpotent Lie algebra, section 3.3 provides one such intuitive and accessible interpretation. The M-layer architecture can be seen as a generalization of this idea to other Lie algebras, and training can demonstrably make it fall back to effectively using a nilpotent algebra to solve some given task, such as computing a determinant. To further address the connection, we have added the following to the paper: "The connection to Lie groups is important because it can provide techniques for the regularization of the M-layer. It is, for example, possible to impose a constraint on the maximal depth of feature crosses by taking a generator-matrix tensor $T_{imn}$ that itself is a tensor product with a nilpotent matrix algebra. There are also other Lie-group-inspired ways to think about regularization, which we will discuss in future work". Considering the space restrictions for this paper and our aim to present here the fundamentals of the M-layer, we did not discuss in more depth regularization, which is tightly linked to Lie theory, but we plan to do this in future work.
> >
> > Finally, the third concern of the reviewer relates to the numerical experiments and comparisons between the M-layer and DNNs. Our paper repeatedly demonstrates that the M-layer is more parameter-efficient than DNNs, while having comparable training speed, which by itself is a good reason to use the former. Regarding experiments involving deeper networks, we had to make a difficult choice about what to include, given the limited space of the paper. As this is a new architecture, we aimed to provide a good explanation of its mathematical foundation, including universal approximation and robustness. We also included experiments showcasing the use of the M-layer both on general ML benchmarks and problems where it has a natural advantage over regular DNNs (domain extrapolation on periodic and geometric data). Rather than going directly to large models where it would be comparatively more difficult to make a proper comparison, we chose an approach with smaller but clearer steps. That being said, we intend to follow up this paper with research on larger models and advanced regularization techniques, including those based on insights from Lie theory. In order to make that more clear, we have added a “Future Work” section.

---

### Official Review · AnonReviewer2 · 2020-10-30
**A nice work with theoretical novelty although experimental study can be further expanded**

**Rating:** 5
**Confidence:** 3

**Review:**

This work proposes a novel machine learning architecture (or M-layer) that is in the form matrix exponential. This architecture can effectively model the interaction of feature components and learn multivariate polynomial functions and periodic functions. The architecture of the M-layer is well described. Its universal approximation capability is explained and proved in the Appendix. Other properties related to periodicity, connection to Lie groups, the interpretation from the perspective of dynamical systems, and the robustness bounds are discussed. Experimental study is conducted on toy datasets and three image classification datasets to demonstrate the properties of the proposed M-layer.

Overall, this is a nice piece of work. The proposed M-layer is novel, and the properties shown by this kind of architecture are interesting, especially the capability in modelling periodic functions and extrapolating data. Theoretical discussion is generally clear although some places are a bit hard to follow. Experimental study supports the claims.

Meanwhile, this work could be further improved at the following points:

1. The experimental study does not combine the M-layer with the convolution layers, and it only tests the M-layer on relatively simple image recognition datasets. It will be interesting to know whether the M-layer can be combined with convolutional layers and trained in an end-to-end manner. This is important to check the potential of the M-layer for the applications of visual, audio, and text data analysis, for which end-to-end training has proven to be more effective. This paper could test this setting on some larger-scale benchmark such as ImageNet if possible.

2. In the experimental study, it will be interesting to see the comparison between the M-layer and Gaussian RBF kernel based SVM (RBF is mentioned in Related Work) on these toy datasets and image benchmarks. RBF-SVM can be regarded as a neural network with one hidden layer.

3. This work indicates that the M-layer can better consider the cross-terms of feature components. This could be related to the recent work on second (or higher)-order visual representation, in which the correlations of the channels in a convolutional feature map are extracted and used as feature representation for the subsequent fully connected and softmax layers. This paper can discuss the potential connections with this line of research ([R1]-[R4])

[R1] Tsung-Yu Lin; Aruni RoyChowdhury; Subhransu Maji, Bilinear CNN Models for Fine-Grained Visual Recognition, 2015 IEEE International Conference on Computer Vision (ICCV).
[R2] Peihua Li; Jiangtao Xie; Qilong Wang; Wangmeng Zuo, Is Second-Order Information Helpful for Large-Scale Visual Recognition? 2017 IEEE International Conference on Computer Vision (ICCV)
[R3] Melih Engin, Lei Wang, Luping Zhou, and Xinwang Liu, DeepKSPD: Learning Kernel-Matrix-Based SPD Representation For Fine-Grained Image Recognition, European Conference on Computer Vision ECCV 2018
[R4] Piotr Koniusz; Hongguang Zhang; Fatih Porikli, A Deeper Look at Power Normalizations, 2018 IEEE/CVF Conference on Computer Vision and Pattern Recognition

4. The capability of "extrapolate learning" of this M-layer is very interesting. This work uses the results in Figures 2 and 3 to demonstrate it. In addition to the double spiral data and the periodic data, is this "extrapolate learning" capability applicable to other more general data or patterns? Please comment.

--- Thank the authors for the detailed response. After reading the response and the comments of peer reviewers, the rating is altered as follows.

---

> ### Author Response · Authors · 2020-11-20
> **We have expanded the baseline and literature comparisons, as requested.**
>
> We thank the reviewer for the valuable comments. We provide below answers to each point.
>
> 1. Combining the M-layer with convolutional architectures is something we plan to explore in future work, as we mention in the newly added “Future Work” section. Due to space constraints, we chose not to include larger experiments involving convolutions and advanced regularization techniques in this paper, but to rather focus on a thorough theoretical explanation. The experiments we included involve tasks that seem relatively simple and might be considered toy problems, but are in fact very hard for DNNs, and clearly show the extrapolation abilities of the M-layer. On the other hand, we chose the experiments on generic benchmarks (image recognition datasets) to demonstrate the M-layer can also perform on par with other *non-specialized* architectures on genetic problems.
>
>
> 2. Note that SVM with RBF kernel has already been studied on the same type of problems, by e.g. Suykens (2001) (see Fig. 5, which demonstrates excellent fitting but the lack of ability to generalize). We nevertheless decided to add RBF-SVM as an additional baseline for all the experiments in our paper - please see the updated figures and table. The results demonstrate that the SVM can learn and generalize very well on data from the same distribution, in multiple cases better than a DNN. However, it is unable to extrapolate beyond the training domain, as best shown in the spiral and the periodicity experiments. In contrast, the M-layer is able to extrapolate such functions.
>
>
> 3. We thank the reviewer for pointing out these references. We have now added and discussed them in Section 2. Specifically, we have emphasized that "the main differences between the M-layer and other architectures that can utilize feature-crosses are the M-layer's ability to also model non-polynomial dependencies (such as a cosine) and the built-in competition-for-total-complexity regularization".
>
>
> 4. Considering its extrapolation properties, the M-layer architecture is expected to be especially useful for regression problems. One specific example where the M-layer is promising is identifying geometric invariants in input data. The determinant is one example of such a geometric invariant. Another example would be the area of a parallelogram spanned by two vectors (and wide generalizations to *d*-dimensional objects embedded in *n*-dimensional space). Given that word embeddings allow us to meaningfully do vector arithmetic (along the lines of "Berlin = Paris - France + Germany"), using an M-layer after an embedding layer could provide more geometric structure to embedding spaces, such as detecting relevance of the volume spanned by three vectors obtained from different aspects of one example by the same trainable embedding. This is a topic that we are exploring in further work.

---

### Official Review · AnonReviewer4 · 2020-10-31
**nice but slightly underwhelming**

**Rating:** 5
**Confidence:** 4

**Review:**

Title: INTELLIGENT MATRIX EXPONENTIATION

Summary of the paper:

The idea is to use the matrix exponential as a layer in a neural network. This basically requires transforming to a latent variable which has a rank two tensor shape (a matrix) then applying the e.g. "expm" function. A robustness result is given (section 3.7) and a universal approximation result (appendix A) and some experiments on mainly toy but also image classification problems, and a univariate regression on seasonal data.

Pros:

The paper is written in a nice easy to read way. I enjoyed reading it because it is very easy to follow, though to be fair the contribution is rather simple so this should be a given.

The robustness result is something, but it follows from a very basic result on matrix exponentials. Also, the universal approximation result is better than nothing, but also quite straightforward and constructive and simply comes from observing the matrix exponential as an arbitrary polynomial.

Section 3 is nice throughout ; simple and easy to follow yet interesting.

Cons:

I hate to be that reviewer, but the paper should go further to motivate the use of the expm function. Figure 2 is very cool indeed, but then the real world experiments are not highly motivating. The seasonality data is neat, but this can presumably be better handled by a Gaussian process with periodic covariance, anyway (which is not compared with). To motivate using expm would require far more extensive experiments, even if this requires some heavy computational resources.

Suggestions:

Figure 1 - mention the need for a matrix layout (you seem to want to suggest a 3 by 3 matrix with your diagram).

Equation 1 - is it normal to omit j and k from the l.h.s. in tensor notation ?

Recommendation:

This paper is crying out for a more thorough experimental evaluation ; I am near the fence on this paper but I think it could be much more impactful with such and as such lean toward rejection.

---

> ### Author Response · Authors · 2020-11-13
> **May seem slightly underwhelming because we chose clear and well-motivated experiments.**
>
> We thank the reviewer for the valuable comments and for the two suggestions on clarifying matrix squareness and the missing tensor indices, which we have now fixed. We were pleased that our paper was easy to follow, as behind the scenes we spent some effort to present clearly multiple concepts that may be unfamiliar to many readers, including Lie groups and the matrix exponential itself. We designed this paper to be an accessible yet thorough introduction to these topics, setting a good theoretical ground for further research on the M-layer.
>
> The straightforward character of the universal approximation proof and certified robustness are, we consider, strengths of the M-layer in terms of interpretability - a key topic in current ML. In the universal approximation proof (which we believe should be mandatory to demonstrate for any new architecture, and not simply "something better than nothing"), the polynomial decomposition is an important insight into the representational capabilities of the model. For certified robustness, we have now expanded on options allowed by the M-layer for future work: "An alternative way of studying robustness for a specific example involves the diagonalization of the matrix that corresponds to that example. In particular, we can use the Baker-Campbell-Hausdorff (BCH) formula to understand the neighborhood of the example. For example, if we regularize the tensor $T$ to punish non-commutativity of generator matrices, we expect the terms in the BCH formula to decrease in magnitude according to the number of commutators, so we can obtain an understanding of the non-linear effects in the neighborhood of each specific example. To our knowledge, no equivalent type of analysis exists for ReLU networks".
>
> Concerning the motivation of using expm, as a side note, this paper originates from a line of unrelated research that extensively used backpropagation through matrix exponentials for sizes up to 1000 x 1000 (which we do not cite in order to preserve anonymity), where we found that this is numerically more efficient than anticipated. We have now added a paragraph to the Introduction, better clarifying the theoretical grounds for this choice: "A unique advantage of the matrix exponential is its natural ability to represent oscillations and exponential decay, which becomes apparent if we decompose the matrix to be exponentiated into a symmetric and an antisymmetric component. [...] This gives the M-layer the possibility to extrapolate beyond its training domain. [...]".
>
> Regarding the experimental evaluation of the M-layer, we had to make a difficult decision about what to explore within the limited space of 8 pages. First, we wanted to provide adequate mathematical background and intuition for readers who would be less familiar with the concepts we used. Then, since this is a new model, we considered it good practice to provide universal approximation and certified robustness proofs. Finally, to relate our model to existing models, we wanted to include its performance on a widely understood generic classification problem, which we deemed the image datasets to be. These experiments demonstrate the validity of the M-layer as a *non-specialized* architecture that can solve generic problems.
>
> In the limited remaining space, we chose to showcase the key properties of the M-layer (domain extrapolation on periodic and geometric data) on relatively simple problems in order to paint a clear picture and avoid confounding factors that might affect larger datasets. We are aware that other techniques can be used to learn in particular periodicity, but, in contrast to these, the M-layer can solve such problems *without any feature engineering*. Importantly, we consider these experiments more relevant to the message of the paper compared to the image recognition experiments. These are not simply "toy problems", but can rather be considered benchmarks for domain extrapolation and parameter efficiency. Conventional DNNs do not perform well on extrapolating even on a simple "Swiss roll" problem, hence the community is not used to regarding these problems from this perspective. We hope this will change.
>
> In summary, we think that our piece of work is valuable and rigorous in accordance to its goal of being a well-motivated mathematical introduction of a new ML architecture with powerful extrapolation properties. Its elegant mathematics should not be confused with conceptual simplicity. In particular, a strength of the M-layer that follows from its mathematical elegance is its potential for interpretability. We chose what to include in the paper with the aim of crafting a clear introduction to the M-layer, as opposed to delving into large experiments without a solid foundation. Nonetheless, we also intend to follow up with research involving more advanced architectures and regularization techniques, in particular some inspired by Lie theory, for larger experiments, as clarified in the “Future Work” subsection.

---

### Decision · Program_Chairs · 2021-01-07
**Final Decision**

**Decision:**

Reject

**Comment:**

This paper was reviewed by 4 reviewers who scored the paper below acceptance threshold even after the rebuttal. Reviewer 4 is concerned about motivation, Reviewer 2 rightly points out that there exist numerous works that use some form of spectral layers in a deep setting on challenging datasets - something lacking in this work. Reviewer 3 is concerned about limited discussion on lie groups and the overall benefit of expm(.). Reviewer 1 reverberates the same comments regarding insufficient experiments,  comparisons and limited motivation. We encourage authors to consider all pointers given by reviewers in any future re-submission.